# Genomic Analysis and Antimicrobial Components of M7, an *Aspergillus terreus* Strain Derived from the South China Sea

**DOI:** 10.3390/jof8101051

**Published:** 2022-10-07

**Authors:** Jiangfeng Qi, Chaoyi Chen, Yajing He, Ying Wang

**Affiliations:** School of Life Science and Technology, China Pharmaceutical University, Nanjing 211100, China

**Keywords:** *Aspergillus terreus*, whole genome sequencing, biosynthesis potentials, antimicrobial compounds

## Abstract

As a typical filamentous fungus, *Aspergillus* species are highly adaptive to diverse ecological habitats, represented by their occurrence in both terrestrial and marine environments; this could plausibly be ascribed to their preeminent biological diversity and metabolic variability. In this context, marine-derived *Aspergillus* fungi have recently attracted great interest as a promising potential source of biologically active compounds. The present study depicts the genomic and chemical profiles of M7, a strain of *Aspergillus terreus* isolated from mussels in the South China Sea; the crude extracts of its soybean fermentation exhibit potent growth-inhibitory properties against *A. baumannii* and *P. aeruginosa*. Subsequently, functional genomics analysis based on sequences implied a considerable biosynthetic potential of the strain, which is substantiated by the 75 biosynthetic gene clusters (BGCs) identified via genome mining; the majority (49 BGCs) were functionally unknown. Representatively, the putative biosynthetic pathways of terramide A and terramide B, the bacteriostatic products obtained through chemical separation and characterized from the fermentation, could not be allocated to any known BGC, highlighting the metabolic potency and diversity of this strain. Meanwhile, based on a comprehensive analysis of fermentation conditions, we confirmed that the presence of environmental iron was inversely correlated with antimicrobial characteristics of the strain M7, presumably due to the interference in the biosynthetic pathway or bioactive mechanisms of the antimicrobial components, e.g., terramide A and B. Our observations provide genomic and biochemical insight into the metabolic and ecological novelties of this strain, underpinning the diversity of biosynthetic flexibility and adaptive strategies of marine *Aspergillus* fungi.

## 1. Introduction 

The ocean covers more than 70% of Earth’s surface and contains a wealth of biodiversity, constituting the largest aquatic ecosystem with tremendous productivity and complexity [1]. The prevalent unfavorable conditions of the marine ecosystem, characterized by high pressure, high salinity, hypoxia, and oligotrophy, have prompted the species that dwell there to develop exclusive genetic and physiological adaptations [2]. Filamentous fungi of marine origin, especially the species of the genera *Aspergillus* and *Penicillium*, are prominent producers of a wide range of secondary metabolites (SMs) with novel scaffolds or valuable bio-functions [3]. Cumulative studies in recent years have demonstrated the diversity, complexity and novelty of marine fungal metabolites, which presumably boost the adaptive plasticity and ecological resiliency of the producing strains, by providing competitive advantages for niche colonization [4]. 

From a general biology perspective, comprehensive and vigorous bioactivities are anticipated for those SMs, which potentially represent an exceptional reservoir to translate into lead candidates or therapeutic agents for disease therapy. Among multiple health challenges, infectious diseases surprisingly remain poised to plague global populations despite the swift development of medical advances. This possibly represents a new era in healthcare, facing the emergence of infection outbreaks due to invasion and transmission of a range of critical superbugs [5]. Given the nature of biosynthesis, microbial SMs serve as competitive weapons that confer adaptive or survival advantages over their rivals, which potentially comprise a fascinating source for the prevention and control of infectious diseases [6]. In our previous work, a variety of filamentous fungi, dominated by *Aspergillus* and *Penicillium* spp., were isolated from sponges and corals in South China, to screen prospective strains with antimicrobial properties [7]. This study delineated the genomic and chemical profiles of a co-epiphytic fungal strain M7 from mussels in the South China Sea, with antimicrobial bioactivities, which subsequently was identified as *Aspergillus terreus*. For comprehensive analysis of the chemical diversity, the M7 strain was subjected to whole genome sequencing for functional annotation of biosynthetic gene clusters (BGCs). The abundance of functional genes insinuated the remarkable metabolic capabilities, presumably conferring on the strain a robust adaptive resilience in the harsh environment, which was represented by a total of 75 putative BGCs. Specifically, among the predicted BGCs, 7 had relatively complete structure and shared high homology with known BGCs, but none were assigned to the synthesis of antimicrobial compounds. Following their antimicrobial properties, terramide A and terramide B were chemically isolated and characterized from fermentation products of the strain on soybean solid medium. Exogenous iron interfered with bacteriostatic action of fermentation products/secondary metabolites in a dose-dependent manner, which might cast some light on the bioactive mechanisms of the SMs, from the perspective of ecological adaptability and metabolic functionality.

## 2. Materials and Methods

### 2.1. Microbial Strains

Strain M7 was isolated from a mussel sample collected on a slope in the South China Sea (17°6′12.2″ N, 111°29′58.6″ E). The purified isolation was inoculated onto a malt extract agar (MEA; OXOID Ltd., Basingstoke, UK) with 3% sea salt, and cultured for 5 days at 28 °C before culture preservation.

The indicator strains *Escherichia coli* ATCC 25922, *Klebsiella aerogenes* ATCC 700603, *Pseudomonas aeruginosa* PA101, methicillin-resistant *Staphylococcus aureus* (MRSA) USA300, methicillin-resistant *Staphylococcus epidermidis* (MRSE) ATCC 35984, *Micrococcus luteus* ACCC 11001, and *Acinetobacter baumannii* ATCC 19606 were stored in the Marine Pharmaceutical Laboratory of China Pharmaceutical University. 

### 2.2. Strain Fermentation for Antibacterial Properties

The strain M7 was activated in the Potato Dextrose Broth (PDB; Solarbio, Beijing, China) from frozen stocks, and then inoculated onto soybean medium (soybean 1000 g, MgSO_4_ 1.5 g, sea salt 65 g, distilled water 1 L) to ferment for 15 days at 28 °C.

Methanol (MeOH) was used to extract the fermentation products, which were then evaporated at reduced pressure to yield a crude extract. The crude extract then was diluted in MeOH at a concentration of 10 mg/mL for bacteriostatic assessment.

### 2.3. Bacteriostatic Assay

Bacteriostatic activity of crude extracts was assessed using agar diffusion method as described with minor modifications [8]. Briefly, 1% of overnight culture of indicator strains was inoculated in Mueller–Hinton Broth (MHB; Solarbio, Beijing, China) and grown at 37 °C and 200 rpm till logarithmic growth stage. 100 μL of bacterial culture (approximately 1 × 10^7^ CFU/mL) was spread on Mueller–Hinton agar (MHA; Solarbio, Beijing, China). Then, 6-mm holes were punched into the agar, which were filled with 30 µL crude extract, standard antibiotics (0.1 mM chloramphenicol) as a positive control, or MeOH as vehicle control. After 24 h statical incubation at 37 °C, the diameters of bacteriostatic rings were measured in millimeters (mm) to indicate the relative bioactivities. The result was expressed by a ratio of inhibitory zone diameter produced by the crude extract to that of positive control, which was defined as strong if the ratio was greater than 1.0, moderate when the scale was between 0.5 and 1, and weak if it was less than 0.5.

### 2.4. Species Identification of M7

For the purified colony of strain M7, morphological analysis was used for preliminary identification. M7 spore suspensions were cultured on MEA for 5 days at 28 °C; macroscopic morphology was observed, in terms of colony characteristics, aerial hyphae type, vegetative hyphae growth, spore formation and pigments excretion. Then, microscopic observation was performed on hyphae and conidiophore structure, followed by lactophenol cotton blue (LPCB) staining [9]. 

Molecular identification of strain M7 was completed by sequencing the internal transcribed spacer (ITS) region of the ribosomal DNA. Genomic DNA was isolated from fresh mycelium of culture grown on MEA. The fungal ITS region was amplified and sequenced with universal primer pairs ITS1 (TCCGTAGGTGAACCTGCGG) and ITS4 (TCCTCCGCTTATTGATATGC). The full-length ITS sequence was sequenced by Sanger sequencing (Applied Biosystems SeqStudio, ABI, Waltham, MA, USA), and analyzed via the Basic Local Alignment Search Tool (BLAST), and aligned using the ClustalW multiple alignment tool with homologous ITS sequences. The phylogenetic tree was then constructed using the neighbor-joining algorithm with 1000 bootstrap replicates in MEGA 7 [10].

### 2.5. Genome Sequencing

Significant bioactivities of the strain fermentation products warranted/justified investigating the genomic/molecular mechanisms responsible for their synthesis. Hence, whole genomes were sequenced on an Illumina Miseq platform by Genewiz Co. (Suzhou, China), as described [11]. For library construction, 100 ng genomic DNA was randomly fragmented to ~500 bp by sonication (Covaris S220, Covaris Inc., Woburn, MA, USA). Paired-end sequencing was performed on the HiSeq platform using HiSeq Control Software. Adapter sequences and low-quality sequences were removed by cutadapt (v1.9.1). Filtered sequence reads were assembled using and gap-filled using Velvet [12], SSPACE [13], and GapFiller [14], respectively. All genome sequencing data have been deposited in the NCBI SRA database (Accession NO: PRJNA848651).

### 2.6. Genome Annotation

Through a homology-based approach, the gene structures were mapped to the reference genome *A. terreus* NIH2624. AUGUSTUS (version 3.3) [15] was used to predict coding genes and high-GC regions. Next, the coding genes were annotated with the NCBI database by BLAST, and the functions of genes were annotated by the Gene Ontology (GO) [16] and Kyoto Encyclopedia of Genes and Genomes (KEGG) [17] databases. In addition, the predicted proteins were classified by the euKaryotic Orthologous Groups (KOG) database [18]. Putative BGCs were predicted on the antiSMASH server through the ClusterFinder algorithm.

### 2.7. Antibacterial Metabolites Separation 

Following fermentation for 15 days at 28 °C, metabolites of strain M7 were extracted by MEOH from soybean cultures containing fungal mycelium. The concentrated crude product using decompressing distillation was subjected to silica gel column chromatography (3 × 60 cm) using a stepwise gradient of CH_2_Cl_2_/ CH_3_OH, yielding 9 fractions (fraction Ⅰ-Ⅸ). Indicated by the antibacterial activity, fraction IV was gradually purified by open ODS, Sephadex LH 20 column chromatography and semi-preparative reversed-phase HPLC (H_2_O-MeOH, 50:50, 2.0 mL/min), yielding active compound 1 and 2.

### 2.8. Antibacterial Metabolites Structural Identification

Compound 1 and 2 both are white solids. We identified their structures by NMR and MS. We dissolved compounds 1 and 2 in CD_3_OD (5 mg/mL) to conduct NMR (Bruker AVANCE 500MHz), and then obtained compounds 1 and 2 ^1^H NMR and ^13^C NMR. The molecular weight of compounds 1 and 2 was identified by a triple quadruple mass spectrometer (Applied Biosystems SCIEX Co, ABI, Waltham, MA, USA).

### 2.9. Effects on the Antibacterial Activities of Exogenous Iron

The potential antibacterial mechanisms of strain M7 were preliminarily investigated by a correlative assessment between the fermentation conditions and bioactivities. A range of concentrations of ferric chloride (0, 1, 2, 3 mg/g) were added to the soybean medium and the strain was grown at 28 °C for 15 days. The fermented cultures were extracted and concentrated as described, to assess antibacterial properties by measuring inhibitory zone diameters (mm). Each sample was represented by three biological replicates.

### 2.10. Statistical Analysis

Statistical analyses were performed in GraphPad Prism by unpaired Student’s *t*-test. Data are expressed as mean ± standard deviation (SD). *p* value < 0.05 was considered statistically significant.

## 3. Results

### 3.1. Antibacterial Activities Screening

The bioactivities of M7 fermented products were evaluated by the growth inhibition against multiple pathogenic bacteria. Compared with standard antibiotics (0.1 mM chloramphenicol), which demonstrated a broad-spectrum activity against Gram-positive (MRSA, MRSE, *M. luteus*) and Gram-negative (*E. coli, A. baumannii, K. aerogenes, P. aeruginosa*) bacteria, the crude extracts of M7 culture inhibit *A. baumannii* and *P. aeruginosa*, specifically. (Figure 1). This indicated that some antibacterial substances were produced during fermentation, apparently being an issue for further study.

### 3.2. Species Identification

Morphological and molecular features were utilized to identify the strain M7. Macroscopically, multiple colony characteristics of the strain were extracted, including shape, appearance, motility, and size. As shown in Figure 2A, the colony on the MEA was white, circular, and floccose, growing prosperously, with a pale and yellow reverse side. Microscopic observations showed that the mycelium formed a long, coarse conidiospore stalk with a subglobose sporangium at its apex, which could be identified as *Aspergillus* sp., as integrated with the macromorphology. 

The full-length ITS region, fungal-specific DNA barcode was amplified and sequenced, with the universal primers (ITS1/ITS4). According to the phylogenetic tree of ITS, the strain M7 was clustered into the *Aspergillus terreus* clade, characterized by the highest identity scores (>99%), which was were distinguished from other species (Figure 2B).

### 3.3. Genome Analysis

To obtain a comprehensive genetic landscape, the whole genome of strain M7 was sequenced with coverage of 211.62×. The draft genome was assembled into an estimated size of 29.2 Mb, with a GC content of 53.11%, and was assembled into 102 scaffolds. The average length of consensus contigs was 368,922.8 bp with an N50 of 1,343,561 bp. From the assembled genome draft, a total of 10,243 proteins were predicted by Augustus software, with an average length of 1565.40 bp (Table 1). 

### 3.4. Genomic Functional Annotation

For genome annotation, the homology-based approach was adopted. The predicted proteins were respectively annotated in the Cluster of Orthologous Groups of proteins (KOG), Kyoto Encyclopedia of Genes and Genomes (KEGG) and Carbohydrate-Active EnZymes (CAZy) databases. 

In the KOG classification, 6048 genes (59.05%) were assigned (Figure 3), which were classified into 26 subcategories of 4 main functional categories: intracellular processes (1582 genes), metabolism (2586 genes), information storage/processing (1147 genes) and poorly characterized function (1495 genes). Among the assumed groups, metabolism appeared as the most prominent one, which involved the transport and metabolism of carbohydrates (374 genes), amino acids (386 genes) and lipids (480 genes), energy production (411 genes), etc. Notably, besides primary metabolism pathways, secondary metabolic pathways were enriched in the genome, hinting at a great possibility of SMs diversity.

Among the allocated 46 KEGG subcategories, the metabolism category was also significantly enriched, of which, secondary metabolism functions were distributed to the biosynthesis, transportation and catabolism of terpenoids, indoles, polyketones, etc. (Figure 4). Similar biological processes described by KOG or KEGG annotation coincidently appeared to provide a genetic basis for the environmental plasticity and adaptation of this strain.

In the CAZy database, 1159 putative proteins were assigned, accounting for 11.31% of the total protein-coding genes. The presumed CAZymes were classified into 6 groups: carbohydrate-binding modules (CBMs), auxiliary activities (AAs), carbohydrate esterases (CEs), polysaccharide lyases (PLs), glycosyl transferases (GTs), glycoside hydrolases (GHs), among which, GHs (39.25%) and GTs (29.42%) demonstrated the most prominent gene abundances (Figure 5).

### 3.5. Prediction of Biosynthetic Potentials

Genes involved in the process of SMs biosynthesis are frequently spatially organized as BGCs in the fungal genome [19]. Following sequence assembly, the biosynthetic potential of strain M7 was probed in antiSMASH, the genome mining platform. As shown in Figure 6A, a total of 75 BGCs were predicted from the assembled 102 scaffolds, with 10 types: beta lactone (2), siderophore (1), indole (4), terpene (6), T1PKS-indole hybrid (1), NRPS-terpene hybrid (1), NRPS-indole hybrid (4), NRPS-T1PKS hybrid (6), T1PKS (22), NRPS (28). In terms of the biosynthetic potentials, the majority of predicted BGCs were functionally unknown, either in M7 (49 BGCs) or in the reference genome (Accession NO: PRJNA15631) of *A. terreus* standard strain NIH2624 (40 BGCs). Of note, there were 9 additional unknown BGCs compared to the reference standard (Figure 6B), which may be supposed to hint at a more robust biosynthetic potential in M7.

Further analysis demonstrated that 7 out of 75 predicted BGCs in M7 genome had relatively complete structure and shared high homology with annotated BGCs in the MIBiG database, implying the possible capacity to produce a range of metabolites, including aspterric acid (Figure 7A), hexadehydroastechrome / terezine-D / astechrome (Figure 7B), squalestatin S1 (Figure 7C), terreic acid (Figure 7D), asperphenamate (Figure 7E), isoterrein (Figure 7F), dihydroisoflavipucine / isoflavipucine (Figure 7G). The presumed biosynthetic pathways and corresponding SMs demonstrated a substantial diversity, in terms of the assembly patterns or chemical structures. As for biological roles, these SMs tended to participate in ecological competition or niche adaptation more than in fungal growth and development. 

For example, aspterric acid could inhibit pollen development in *Arabidopsis* [20]; terreic acid has antibiotic properties, and it can act against various Gram-positive and Gram-negative bacteria through inhibition of the bacterial cell wall biosyntheses [21]; squalestatin 1 is a potent inhibitor of squalene synthase, which lowers serum cholesterol in vivo [22].

### 3.6. Chemical Isolation of Metabolites

Intriguingly, functional analysis failed to reveal the antimicrobial potentials of the SMs inferred by genome mining, which highlighted the necessity of chemical separation. Directed by the bacteriostatic bioactivities against *A. baumannii*, compound 1 and 2, were isolated and characterized from soybean cultures of strain M7, by sequential column chromatography (Figure 8). The molecular mass of compound 1 was determined by LRESI-MS [M-H]^−^ at m/z 257.3 (Appendix A), and together with ^1^H NMR and ^13^C NMR (Tanble S1); the molecular mass of compound 2 was determined by LRESI-MS [M-H]^−^ at m/z 243.1 (Appendix A), and together with ^1^H NMR and ^13^C NMR (Tanble S2). This indicated that the compound 1 was terramide A and compound 2 was terramide B (Figure 8). Both chemicals are white amorphous powders, with a MIC of 32 µg/mL (Terramide A) and 64 µg/mL (Terramide B) against *A. baumannii* ATCC 19606, respectively (data reported in another work).

### 3.7. Effects of Exogenous Iron on Bioactivities of Crude Extracts

Structure-function analysis suggested that both terramide A and B contain hydroxamate moieties, implying a potential for them to function as fungal siderophores, given that siderophores are crucial for iron uptake and are enhanced under iron-limiting conditions. It is plausible to speculate that iron might implicate in biological function of those SMs, which might provide insight into the underlying mechanisms. To verify this, a range of concentrations of ferric chloride (0, 1, 2, 3 mg/g soybean medium) were added during fermentation. According to the inhibition zones against *A. baumannii* ATCC 19606, the bacteriostatic potency of fermentation products was negatively correlated with exogenous iron, and was completely blocked at higher concentration (3 mg/g, Figure 9).

## 4. Discussion

*A. terreus* is prevalent in both terrestrial and marine habitats due to its outstanding environmental adaptability [23], implying a genetic abundance and metabolic diversity. The wide distribution in harsh niches should testify to the survival privilege or colonization fitness of the species, presumably conferred by various functional SMs, possibly with antimicrobial, antiparasitic, fungicidal and herbicidal effects, which could be developed into lead compounds in pharmaceutical, agricultural, and industrial fields. In recent decades, *A. terreus* has become a prolific producer of various natural products with bioactive properties and diverse applications, e.g., lovastatin, sulochrin, terrein, and itaconic acid [24,25]. 

Strain M7, a filamentous fungus, was isolated from mussels in the South China Sea, and identified as *A. terreus* by phenotype and phylogeny in this study. Following preliminary fermentation and product extraction, the bioactivity evaluation of the strain exhibited a promising result against *A. baumannii* and *P. aeruginosa*. Subsequently, the whole genome of this strain was sequenced and genome-based functional analysis was performed, to provide a genetic map of its metabolic and adaptative characteristics. 

Genome analyses revealed an abundance and diversity of functional genes, of which, metabolism-related genes were coincidentally enriched in multiple of databases. For instance, metabolism category constituted the primary part in either the KEGG or KOG database; and SMs biosynthesis represented a substantial fraction (6.51%) of KOG functional subcategories. Meanwhile, putative CAZymes contributed 11.31% of the total protein-coding genes against the CAZy database. Those results were integrated to indicate a genetic abundance of this strain, at least in the metabolic range, expressly for primary and secondary metabolites.

The metabolic diversity was substantiated by genome mining, which identified a total of 75 putative BGCs. Among the 10 types of BGCs, T1PKS and NRPS provided 66.7% of the total biosynthetic pathways, as well as multiple NRPS-T1PKS hybrids (6 clusters, 8.0%), highlighting the dependence of biosynthesis on those mechanisms. In terms of structural or functional analogy, a considerable proportion of predicted BGCs in the M7 strain remain elusive, possibly due to limited annotations of fungal genes, as was the case even within the best understood genome of *A. terreus* (standard strain NIH2624) [26]. However, the M7 genome included 9 unknown BGCs that exceeded the reference genome, which, in the genetic context, might represent a greater richness of metabolites, and an exciting future challenge as well. Presumably, given its oceanic origin, the ecological fitness explains the robust metabolic pattern, having bestowed an adaptive plasticity to this strain which surely needs further investigation [27].We find a complete BGC for aspterric acid within the M7 genome (Figure 10), consisting of a core enzyme (AstA) and relevant modification enzymes (AstB-D). More specifically, trichothecene synthase (AstA), cytochrome P450 oxidases (AstB, AstC), and a dihydroxy acid dehydratase (AstD) are involved in the construction of the trichothecene backbone and subsequent modification processes [28]. As a carotene-type sesquiterpene first identified from *A. terreus*, aspterric acid is a natural herbicide that inhibits pollen development, reduces stem length, and retards bolting and first flowering in *A. thaliana* [20]. Moreover, it has been observed that this metabolite promotes the transdifferentiation of pancreatic islet alpha cells into beta cells in eukaryotes [29].

In contrast, functional annotation failed to establish a relationship between predicted metabolites and the antibacterial activity of the strain, necessitating chemical separation and characterization of fermentation cultures. Through bioactivity-guided fractionation, terramide A and B were obtained as antibacterial components; this biological activity was rarely observed in previous reports [30]. Given the isohydroxamate moieties present in the structural skeleton, the metabolites could act as fungal siderophores [31], implying the involvement of iron; this was substantiated by a correlational study between exogenous iron and the antibacterial activity of the fermentation products. As the external iron increased, the antibacterial property declined until it was eliminated, which may be attributable to a disruption in metabolite production or action mode, an issue for further investigation.

## 5. Conclusions

In the present study, a strain of *A. terreus*, M7, was isolated from mussels in the South China Sea. Its biosynthetic capacity was holistically profiled through a combination of genome mining and chemical isolation. The significant richness and diversity of functional genes were revealed by whole genomic analysis, although most of them remain unannotated. This would indicate a metabolic flexibility and viability in niche adaptability for this strain. From the fermentation extracts, chemical separation identified terramide A and B as the antibacterial agents, the activities of which may be entangled with environmental iron. Our data provide additional insights into the genomic and biochemical characteristics of *A. terreus*, especially maritime strains, which apparently would facilitate the understanding and exploitation of these biomasses, whether in the perspective of biology or chemistry.

## Figures and Tables

**Figure 1 jof-08-01051-f001:**
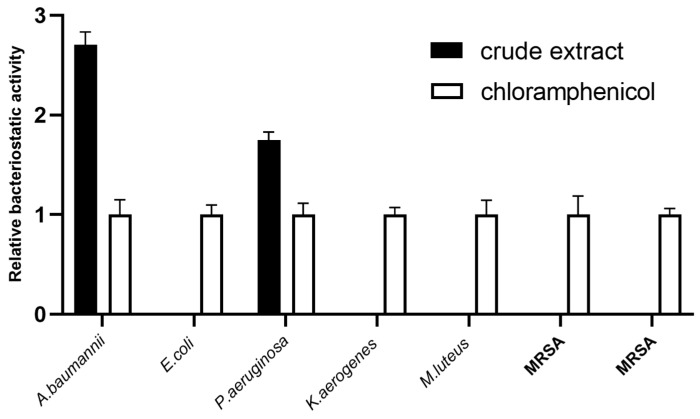
The antibacterial activities of M7 soybean crude extract. The antibacterial activity of the crude extract was indicated by the ratio of the inhibitory zone produced by crude extract to that of positive control (0.1 mM chloramphenicol). The white bar indicated as 1, the black indicated as the ratio of the inhibition zone relative to that of the positive control.

**Figure 2 jof-08-01051-f002:**
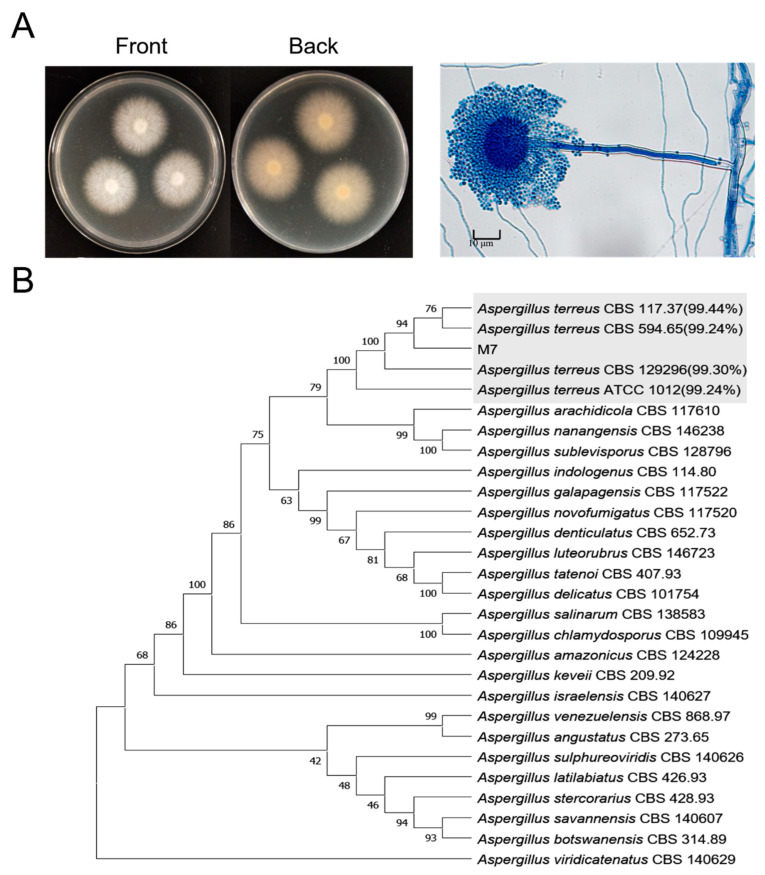
Species identification of strain M7. (**A**) Morphology of M7. Colony, and microscopic morphology were observed after 5 days of incubation on MEA at 28 °C. Morphological features were demonstrated for obverse, reverse side of colonies, conidiophores and conidiogenous cells (400 × magnification; scale bar 10 µm). (**B**) Phylogenetic tree of strain M7 based on the full length of ITS sequences. Phylogenetic analyses were constructed by neighbor-joining algorithm in MEGA7 with 1000 bootstrap replications. The shaded part contained *Aspergillus terreus*, and noted their similarity to M7.

**Figure 3 jof-08-01051-f003:**
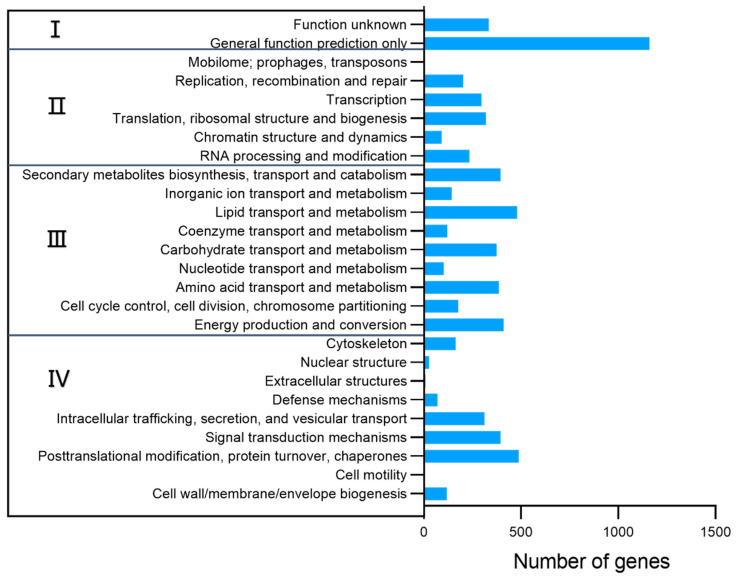
KOG classifications of putative proteins in the genome of M7. I: poorly characterized function; II: information storage/processing; III: metabolism; IV: intracellular processes.

**Figure 4 jof-08-01051-f004:**
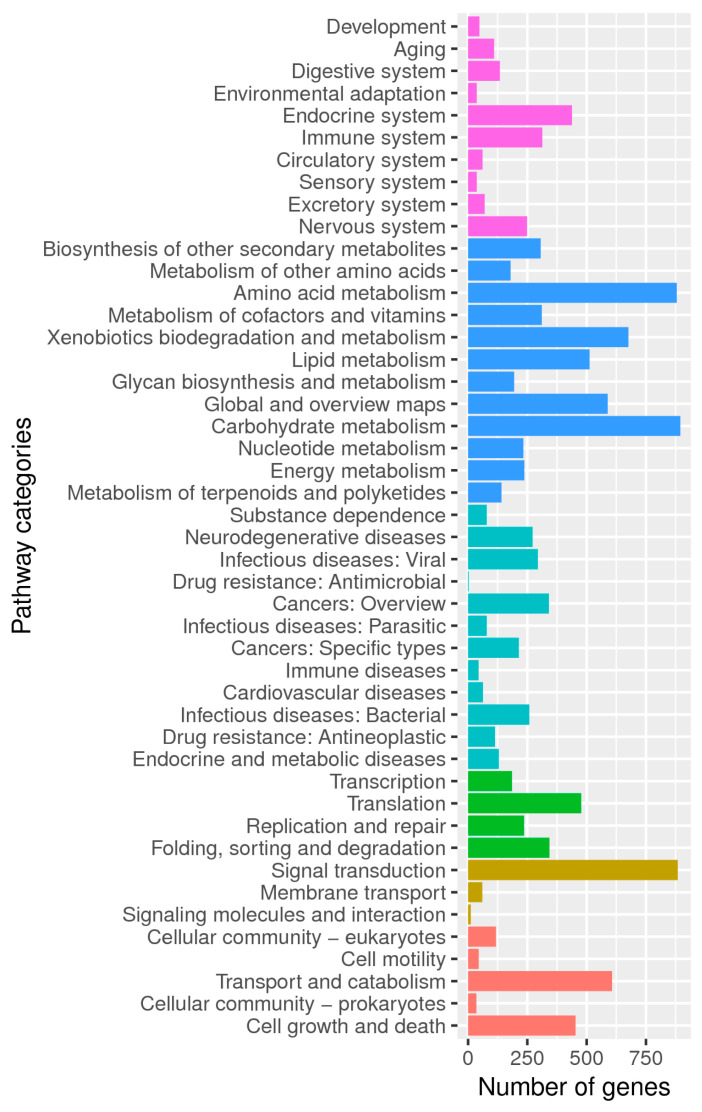
KEGG classifications of predicted coding genes in the genome of M7. Different functional class was represented by a unique color: red, Cellular Processes; brown: Environment Information Processing; green: Genetic Information Processing; cyan: Human Diseases; blue: Metabolism; purple: Organismal Systems.

**Figure 5 jof-08-01051-f005:**
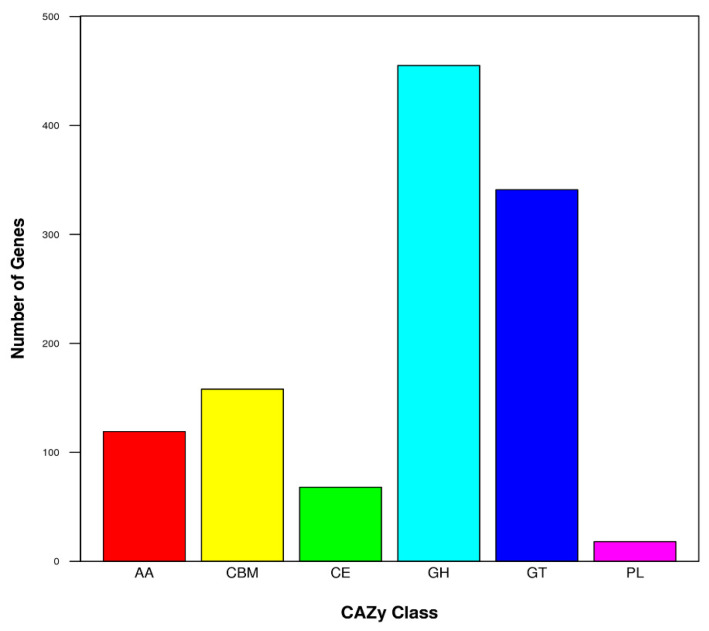
CAZy classifications of putative proteins in the genome of M7. AA, auxiliary activities; CBM, carbohydrate-binding modules; CE, carbohydrate esterases; PL, polysaccharide lyases; GT, glycosyl transferases; GH, glycoside hydrolases.

**Figure 6 jof-08-01051-f006:**
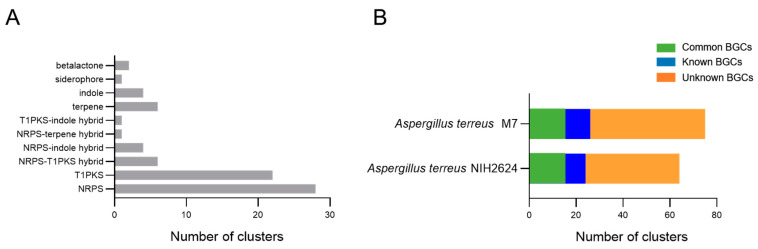
BGCs predicted in M7 genome. (**A**) Biosynthetic types and distribution of putative BGCs in M7 genome. (**B**) Comparison of the predicted BGCs in M7 genome with those in *A. terreus* reference genome (Accession NO: PRJNA15631) of *A. terreus* standard strain NIH2624. NRPS, nonribosomal peptide synthase; T1PKS, Type I Polyketide Synthase; polyketide synthase.

**Figure 7 jof-08-01051-f007:**
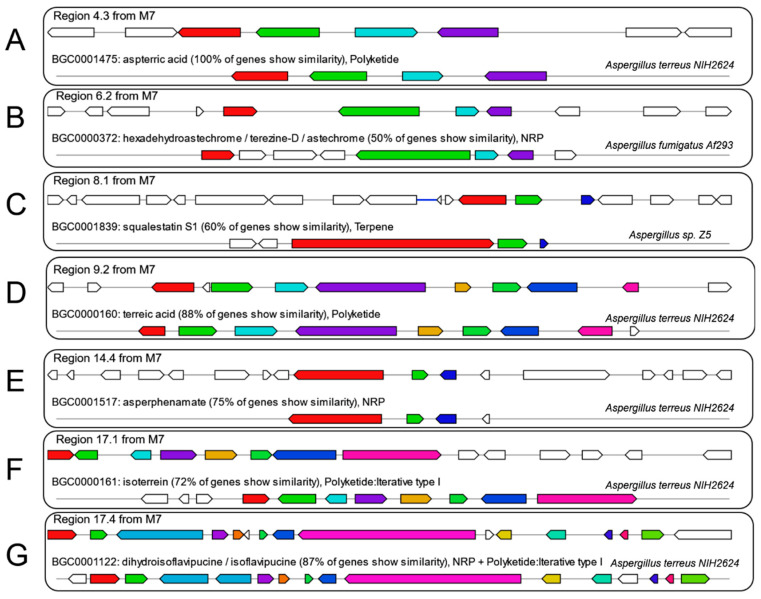
Schematic representation of the putative BGCs and homologous BGCs in the M7 genome and MIBiG database. (**A**–**G**) The upper part represents putative BGC in M7, followed by the characterized BGCs in the MIBiG database.

**Figure 8 jof-08-01051-f008:**
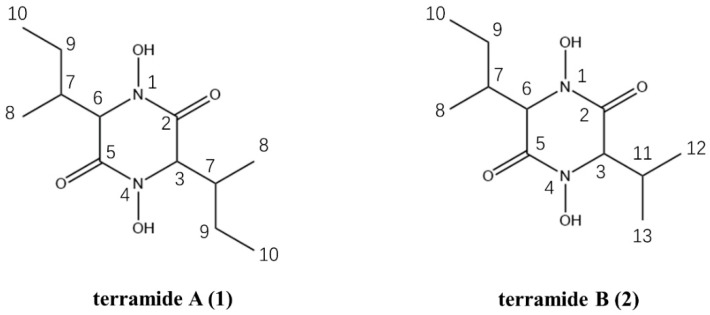
Terramide A (1) and terramide B (2) isolated from fermentation products of M7.

**Figure 9 jof-08-01051-f009:**
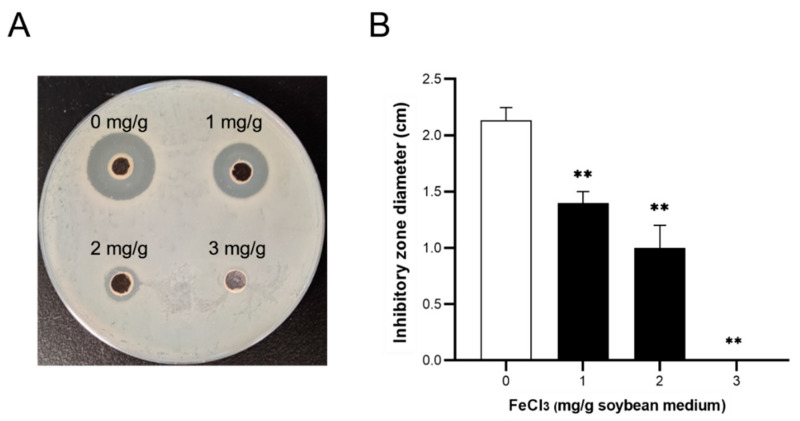
The exogenous iron dose-dependently affects the antibacterial activities of fermentation products. (**A**) Respective image of inhibitory zones altered by exogenous iron. Ferric chloride was added at the indicated concentrations (0, 1, 2, 3 mg/g soybean medium) during fermentation, and then the inhibitory activities of fermentation products were evaluated. Inhibitory potencies were assessed by the diameters (cm) of the inhibition zone against *A. baumannii* ATCC 19606 using the plate diffusion method. (**B**) Data represented the means ± SD of triplicate biological replicates (*n* = 3); **: *p* < 0.01, denoted significant difference vs. control (0 mg/g ferric chloride added).

**Figure 10 jof-08-01051-f010:**
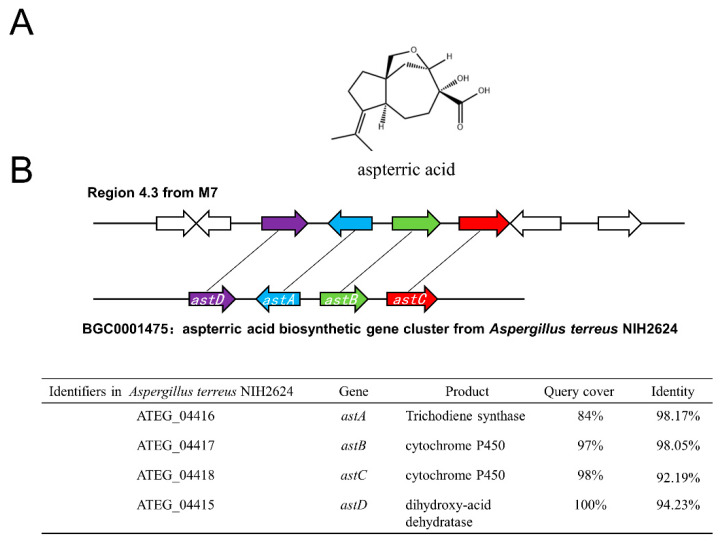
Aspterric acid deduced by putative BGC in strain M7. (**A**) Chemical structure of aspterric acid. (**B**) Schematic comparison of putative BGC of aspterric acid between A7 origin and standard strain origin. Homologous genes in Region 4.3 and the known BGC from the reference genome (Accession NO: PRJNA15631) are colored identically. Accordingly, BGC descriptions and amino acid homology (query cover and identity) are listed.

**Table 1 jof-08-01051-t001:** General features of the M7 genomes.

Genome	Value
Assembly size (Mb)	29.2
G+C (%)	53.11
Assembled scaffolds	102
N50 length (bp)	1,343,561
Average length (bp)	286,659.1
Predicted protein-coding genes	10,243
The average length of predicted protein-coding genes (bp)	1565.40
The average depth of reads cover	211.62
Sequencing method	Illumina HiSeq

## Data Availability

Not applicable.

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
