# Peer review of "Genomic Analysis and Antimicrobial Components of M7, an Aspergillus terreus Strain Derived from the South China Sea"

_jof, 2022, doi:10.3390/jof8101051_

Round 1
Reviewer 1 Report
In Qi et al, the authors have isolated a strain of filamentous fungi from mussels in the South China Sea. They isolated small molecules produced by the organism during fermentation and tested them for bioactivity against a panel of bacteria including Gram-positive and Gram-negative species with relevance to human health. Using the disc diffusion method, this study found that they fungal extract produced significant inhibition against Acinetobacter baumannii and Pseudomonas aeruginosa. This result prompted further investigation culminating in WGS, revealing the organism to be Aspergillus terreus M7. Bioinformatic analysis was used to partially annotate the genome, and it was revealed that the A. terreus M7 encodes 75 putative BGCs. Further investigation into the bioactivity of the crude metabolite extract revealed that activity was attributable to two compounds, terramide A and terramide B. These compounds harbour chemical moieties associated with siderophores, leading to the authors investigating the role of iron on production of these compounds under their fermentation conditions. The authors found that increasing bioavailable iron during growth resulted in lower levels of bioactivity in crude metabolite extracts, seemingly corroborating the hypothesis that terramide A and terramide B may be functioning as siderophores and produced in response to low iron availability.
Required Revisions
1. Key methodological details related to NMR-determination of the structure of terramide A and terramide B are missing. There should be a section added to the Materials and Methods that outlines how NMR was conducted.
2. Figure 7 is unclear. For the representative clusters from MIBiG database, only the backbone enzymes are depicted with the exception of the aspterric acid cluster. Furthermore, it is not clear was meant by the phrase “100% of genes show similarity”. This phrase is used in each example and could be interpreted as the genes in M7 sharing 100% identity with the reference BGC. This is clearly not the case, so this should be reworded to avoid confusion. Figure 7C is particularly problematic, this looks to be a truncated and non-functional PKS but it presented as a putatively active cluster with homology to fusarin BGCs. This discrepancy could be further investigated by including information on accessory enzymes within this BGC, but only the backbone enzyme is shown. Finally, in all cases, the host organism of the reference clusters used should be included in the figure.
3. Figure 8 – the reference genome containing the aspterric acid cluster used for comparison with M7 should be included. Without data to support production of aspterric acid by M7, such as mass spectrometry data, this figure might be better included within the discussion section.
Minor Revisions
1. The BGC responsible for terramide biosynthesis is not proposed. The authors present interesting results related to the effect of iron availability on terramide production, but do not present follow up experiments to further investigate this observation. If extracellular iron abrogates terramide production, it is possible that this is due to modulation of terramide BGC expression. This is something that could be readily investigated using RNA-seq transcriptomics. It may be beyond the scope of the presented manuscript but does strike me as a glaring omission in the context of the larger study, which used bioactivity as a rationale for genomic investigation.
Typos and minor corrections
Line 48: “expectable” should be changed to “anticipated”
Line 54: “privileges” should be changed to “advantages”
Author Response
Response to reviewers
We thank the reviewers for their constructive and helpful suggestions. Our responses to the reviewers’ comments are provided below (with Reviewers’ comments in italic).
Required Revisions
- Key methodological details related to NMR-determination of the structure of terramide A and terramide B are missing. There should be a section added to the Materials and Methods that outlines how NMR was conducted.
We apologize for the omission of details determination of the structure of terramide A and terramide B. We have added the corresponding NMR and mass spectrometry data in the chapter Materials and Methods.
- Figure 7 is unclear. For the representative clusters from MIBiG database, only the backbone enzymes are depicted with the exception of the aspterric acid cluster. Furthermore, it is not clear was meant by the phrase “100% of genes show similarity”. This phrase is used in each example and could be interpreted as the genes in M7 sharing 100% identity with the reference BGC. This is clearly not the case, so this should be reworded to avoid confusion. Figure 7C is particularly problematic, this looks to be a truncated and non-functional PKS but it presented as a putatively active cluster with homology to fusarin BGCs. This discrepancy could be further investigated by including information on accessory enzymes within this BGC, but only the backbone enzyme is shown. Finally, in all cases, the host organism of the reference clusters used should be included in the figure.
This is a fair and accurate comment. At first, we only focused on the similarity of single gene between the predicted gene cluster in M7 and the reference gene cluster, and did not consider the overall similarity of the whole gene cluster. We have re-analyzed the predicted gene clusters, including more complete gene cluster information.
- Figure 8 – the reference genome containing the aspterric acid cluster used for comparison with M7 should be included. Without data to support production of aspterric acid by M7, such as mass spectrometry data, this figure might be better included within the discussion section.
We agreed with the suggestion that figure 8 should be included in the discussion section and we have re-edited this section. In this section, we used Aspergillus terreus NIH2624 (Accession NO: PRJNA15631) as reference genome to compare with M7.
Minor Revisions
- The BGC responsible for terramide biosynthesis is not proposed. The authors present interesting results related to the effect of iron availability on terramide production, but do not present follow up experiments to further investigate this observation. If extracellular iron abrogates terramide production, it is possible that this is due to modulation of terramide BGC expression. This is something that could be readily investigated using RNA-seq transcriptomics. It may be beyond the scope of the presented manuscript but does strike me as a glaring omission in the context of the larger study, which used bioactivity as a rationale for genomic investigation.
The biosynthetic gene cluster of Terramide has not been identified so far, which is the main reason why we could not predict its BGC. Using RNA-seq transcriptomics to investigate the expression of terramide BGC is a good suggestion, and it will be added to our subsequent research.
Typos and minor corrections
Line 48: “expectable” should be changed to “anticipated”
Line 54: “privileges” should be changed to “advantages”
We are sorry for the negligence, and we have made the revisions.
Reviewer 2 Report
The manuscript is a good set of experiments that detect and characterize the antimicrobial activity of terramides produced by a marine Aspergillus isolate. The authors sequenced the genome of an A. terreus isolated from mussels, adding a marine sample to the ~15 published A. terreus genomes. It would have been interesting to see the M7 genome compared to more of the published terreus genomes, but this appears to have been beyond the scope of this study. It would have also been interesting to know if the changes in metabolite activity on preventing microbe growth in response to iron was indeed a result of reduced synthesis of terramides. However the work is still an adequate contribution and paves the way for the other analyses just mentioned. I offer these comments in a collegial spirit with the hope of improving the manuscript's clarity, which would hopefully serve to broaden the study's impact. I consider these comments minor, hoping that they would inform minor revisions to the manuscript. Thank you. Line 42: Since the manuscript integrates both genomic and chemical methods, define "scaffold" here as chemical scaffold (my assumed meaning) vs. genomic scaffold (i.e. assembled contigs). Lines 69-73: Consider breaking this sentence into multiple sentences and making the points more explicit. "Exogenous iron is supposed to implicate in the bacteriostatic actions via investigation on the fermentation conditions, plausibly as a result of its perturbation of the generation or action processes of the antimicrobial compounds," might be reworded to something like "Exogenous iron interfered with bacteriostatic action of fermentation products/secondary metabolites in a dose-dependent manner." ",which might cast some light on the bioactive mechanisms of the SMs, from the perspective of ecological adaptability and metabolic functionality." consider rephrasing. For me personally rephrasing this sentence would make it easier to understand the main findings of your study before diving into the methods. Lines 112-119: Please specify how the fungal ITS region was sequenced (i.e. what method / machine) to improve replicability. Line 121: Consider rephrasing this sentence. "Significant bioactivities of the strain fermentation products underscored the essentials of further investigation from a biological aspect." could be reworded to something like "Significant bioactivities of the strain fermentation products warranted/justified investigating the genomic/molecular mechanisms responsible for their synthesis." Line 138: antiSMASH, not "an-tiSMASH". please include version numbers for software used to help with reproducibility. Line 141: Remove "The" : "Following fermentation..." Line 148: Please add information about the NMR machine (i.e. brand, model). Line 158: add space before "Data..." Line 167: If by "significantly" you mean statistically significant, please include results of statistical analyses, at least a P-value. Line 169 Figure 1: Typically bars in a bar chart represent data. For that reason, rather than including a gray bar marking 1 for each treatment, why not simply include a vertical line marking 1. Also, presumably no orange bar is shown for five of the treatments because the ratio of the inhibitory zone produced by crude extract to that of the positive control is negative. Please consider showing these data by extending the response variable axis (in this case the horizontal axis) to be negative. Also since this experiment was performed with three replicates per treatment please consider adding data points for each replicate, and perhaps including error bars. Line 371: "most of them remain [functionally] unnannotated." I find it odd that the authors do not cite as reference the 2021 study (first author is Ryngajłło) "Complete genome sequence of lovastatin producer Aspergillus terreus ATCC 20542 and evaluation of genomic diversity among A. terreus strains." published in Applied Microbiology and Biotechnology In the discussion, consider addressing the question: How many BGCs are typical of an Aspergillus genome? And how your discovery of 75 BGCs compares to this number. Line 360: "terramide A and B ... rarely observed in previous reports" I find this statement confusing because terramide A, B, and C were characterized as products of A. terreus in 1986 by Garson MJ et al. in J. Chem. Soc. Are they actually all that uncommon across A. terreus isolates?
Author Response
Response to reviewers
We thank the reviewers for their constructive and helpful suggestions. Our responses to the reviewers’ comments are provided below (with Reviewers’ comments in italic).
Line 42: Since the manuscript integrates both genomic and chemical methods, define "scaffold" here as chemical scaffold (my assumed meaning) vs. genomic scaffold (i.e. assembled contigs).
We are sorry that “scaffolds” here confused the readers. But in combination with the previous secondary metabolites (SMs), the definition of “scaffolds” is chemical scaffold.
Lines 69-73: Consider breaking this sentence into multiple sentences and making the points more explicit. "Exogenous iron is supposed to implicate in the bacteriostatic actions via investigation on the fermentation conditions, plausibly as a result of its perturbation of the generation or action processes of the antimicrobial compounds," might be reworded to something like "Exogenous iron interfered with bacteriostatic action of fermentation products/secondary metabolites in a dose-dependent manner." ",which might cast some light on the bioactive mechanisms of the SMs, from the perspective of ecological adaptability and metabolic functionality." consider rephrasing. For me personally rephrasing this sentence would make it easier to understand the main findings of your study before diving into the methods.
We deeply appreciate the detailed and throughout reviews on the manuscript, and we have made changes to this.
Lines 112-119: Please specify how the fungal ITS region was sequenced (i.e. what method / machine) to improve replicability.
This is a fair and accurate comment. Following the suggestion, we added sequencing method (Sanger sequencing) and machine (Applied Biosystems SeqStudio).
Line 121: Consider rephrasing this sentence. "Significant bioactivities of the strain fermentation products underscored the essentials of further investigation from a biological aspect." could be reworded to something like "Significant bioactivities of the strain fermentation products warranted/justified investigating the genomic/molecular mechanisms responsible for their synthesis."
We think this is a good proposal and we have amended it
Line 138: antiSMASH, not "an-tiSMASH". please include version numbers for software used to help with reproducibility.
We are sorry for the negligence. Following the suggestion, we have amended it.
Line 141: Remove "The" : "Following fermentation..."
We have made the revisions.
Line 148: Please add information about the NMR machine (i.e. brand, model).
This is a fair and accurate comment. Following the suggestion, we added information of the NMR machine (Bruker AVANCE 500MHz).
Line 158: add space before "Data..."
We have made the revisions.
Line 167: If by "significantly" you mean statistically significant, please include results of statistical analyses, at least a P-value.
Line 169 Figure 1: Typically bars in a bar chart represent data. For that reason, rather than including a gray bar marking 1 for each treatment, why not simply include a vertical line marking 1. Also, presumably no orange bar is shown for five of the treatments because the ratio of the inhibitory zone produced by crude extract to that of the positive control is negative. Please consider showing these data by extending the response variable axis (in this case the horizontal axis) to be negative. Also since this experiment was performed with three replicates per treatment please consider adding data points for each replicate, and perhaps including error bars.
Thanks for the meticulous and professional reviews on our paper, and we have redrawn Figure 1 as suggested.
Line 371: "most of them remain [functionally] unnannotated." I find it odd that the authors do not cite as reference the 2021 study (first author is Ryngajłło) "Complete genome sequence of lovastatin producer Aspergillus terreus ATCC 20542 and evaluation of genomic diversity among A. terreus strains." published in Applied Microbiology and Biotechnology.
We appreciate this fair and constructive suggestion. The “most of them remain unannotated” here referred to predicted BGCs by antiSMASH, and which can be annotated as PKS, NRPS, etc., but no information about specific compounds.
In the discussion, consider addressing the question: How many BGCs are typical of an Aspergillus genome? And how your discovery of 75 BGCs compares to this number.
This is a fair and accurate comment. The number of BGC in Aspergillus is still quite different, so we chose A. terreus NIH2624 similar to M7 as a reference to compare the number of BGCs.
Line 360: "terramide A and B ... rarely observed in previous reports" I find this statement confusing because terramide A, B, and C were characterized as products of A. terreus in 1986 by Garson MJ et al. in J. Chem. Soc. Are they actually all that uncommon across A. terreus isolates?
We are sorry for the negligence. What we're trying to say here is that the antibacterial biological activity of “terramide A and B” is rarely observed in previous reports. So we have made the revisions.
Round 2
Reviewer 1 Report
The authors have adequately addressed my previous comments.